# Avoiding 'second victims' in healthcare: what support do staff want for coping with patient safety incidents, what do they get and is it effective? A systematic review

Ruth Simms-Ellis ,[1,2] Reema Harrison ,[3] Raabia Sattar,[1] Elizabeth Sweeting,[4] Hannah Hartley,[1] Matthew Morys-Edge,[2] Rebecca Lawton [1,2]

[1]The Yorkshire Quality and Safety Research Group, Bradford Institute for Health Research, Bradford, UK
[2]School of Psychology, University of Leeds, Leeds, UK
[3]Australian Institute of Health Innovation, Macquarie University, Sydney, New South Wales, Australia
[4]Improvement Academy, Bradford Institute for Health Research, Bradford, UK

**Correspondence to**
Dr Ruth Simms-Ellis;
r.simms-ellis@leeds.ac.uk

## ABSTRACT

**Objectives** Incontrovertible evidence surrounds the need to support healthcare professionals after patient safety incidents (PSIs). However, what characterises effective organisational support is less clearly understood and defined. This review aims to determine what support healthcare professionals want for coping with PSIs, what support interventions/approaches are currently available and which have evidence for effectiveness.

**Design** Systematic research review with narrative synthesis.

**Data sources** Medline, Scopus, PubMed and Web of Science databases (from 2010 to mid-2021; updated December 2022), reference lists of eligible articles and Connected Papers software.

**Eligibility criteria for selecting studies** Empirical studies (1) containing information about support frontline healthcare staff want before/after a PSI, OR addressing (2) support currently available, OR (3) the effectiveness of support to help prevent/alleviate consequences of a PSI. Study quality was appraised using the Quality Assessment for Diverse Studies tool.

**Results** Ninety-nine studies were identified. Staff most wanted: peer support (n=28), practical support and guidance (n=27) and professional mental health support (n=21). They mostly received: peer support (n=46), managerial support (n=23) and some form of debrief (n=15). Reports of poor PSI support were common. Eleven studies examined intervention effectiveness. Evidence was positive for the effectiveness of preventive/preparatory interventions (n=3), but mixed for peer support programmes designed to alleviate harmful consequences after PSIs (n=8). Study quality varied.

**Conclusions** Beyond peer support, organisational support for PSIs appears to be misaligned with staff desires. Gaps exist in providing preparatory/preventive interventions and practical support and guidance. Reliable effectiveness data are lacking. Very few studies incorporated comparison groups or randomisation; most used self-report measures. Despite inconclusive evidence, formal peer support programmes dominate. This review illustrates a critical need to fund robust PSI-related intervention effectiveness studies to provide organisations with the evidence they need to make informed decisions when building PSI support programmes.

**PROSPERO registration number** CRD42022325796.

## STRENGTHS AND LIMITATIONS OF THIS STUDY

⇒ We applied a respected multilevel occupational health stress prevention model to patient safety incidents and used this model to map second victim support requirements and organisational support provision.
⇒ We specifically targeted intervention effectiveness.
⇒ Data extraction, quality assessment and narrative synthesis were undertaken independently by two reviewers to address potential issues of subjective judgement and lack of transparency.
⇒ Our reporting was guided by the synthesis without meta-analysis framework.
⇒ Our search strategy may have omitted eligible studies, given the complexity of the research area, our focus on papers published in English between 2010 and 2022 and our selection and number of bibliographic search engines.
⇒ Our final sample comprised studies that were heterogeneous in terms of quality, design, sample sizes, data description and analysis and outcomes measured.

## INTRODUCTION

The international healthcare workforce is in crisis,[1] with an estimated global deficit of 9.9 million nurses, midwives and physicians by 2030.[2] High turnover is a major contributor[3 4] and this has accelerated during and since the COVID-19 pandemic.[5] Being involved in a 'patient safety incident' (PSI) is a significant factor in healthcare staff turnover[6–8]: around 12% of healthcare professionals surveyed in one study had left their job after a PSI.[9]

PSIs are unintended or unexpected incidents, which harmed or could have harmed

one or more patients receiving healthcare.[10] PSIs occur in around 10%[11] to 14%[12] of patient admissions in hospital and in 2%–3% of patient visits in primary care.[13 14] Being involved in harming or almost harming a patient can be traumatising for healthcare professionals, who typically enter healthcare to help others and uphold the Hippocratic oath of 'first do not harm'.[15] This human distress is frequently compounded and intensified by harsh retributive blame culture,[16–21] a judgemental or critical response from colleagues and supervisors, inadequate support in disclosing what happened to the patient/family, the subsequent investigation process[22] and fears about litigation and malpractice.[15]

The first priority after any PSI is to care for the patient and family, as the primary victims. However, there is incontrovertible evidence that healthcare professionals involved in such incidents also require their organisation's support as the silent 'second victims'.[23] With a high number of healthcare professionals across different roles and countries reporting having experienced the second victim phenomenon at some point in their career (recent surveys range from 45% of 205 US obstetrics and gynaecology workers[24] to 91% of 658 Asian surgeons)[25], the scale and universality of this issue is significant.

The potential, harmful impact of PSIs on individual healthcare professionals and healthcare organisations has been well documented through over 20 years of international research into the second victim phenomenon.[6 7 15 16 26–35] Individuals consistently describe experiencing a range of emotional, cognitive, behavioural and physiological symptoms (table 1). Many of these symptoms are commensurate with a clinical diagnosis for post-traumatic stress disorder (PTSD)[36] and burnout, which is significantly associated with involvement in a PSI.[34 37] Impacts on professional functioning include an impaired ability to manage patients in the aftermath of a PSI, reduced confidence, self-doubts, fears about litigation, losing one's job,[21 34] defensive changes in practice, such as avoiding procedures, secrecy/cover ups and ordering up to 30% more tests than necessary.[38] The organisational impact of PSIs include compromised team functioning, patient safety and care quality,[34] increased costs from defensive practice and reluctance to report future incidents,[38] absenteeism, turnover and the loss of highly trained staff.[6]

While there is incontrovertible evidence for the need to support healthcare professionals with PSIs, less clarity surrounds what type of support they actually want and tend to receive or what characterises an 'effective' support intervention.[39] A recommended model for supporting healthcare professionals *after* involvement in PSIs was developed inductively by Scott *et al*,[40] featuring three tiers: local support from leaders and colleagues (tier 1); well-being support from trained peers and patient safety colleagues (tier 2); a referral network for staff requiring more help (tier 3). A subsequent early review by Seys *et al*[41] of the literature surrounding second victim support corroborated this multilevel approach. Their recent, updated review[42] also recommended incorporating an additional 'prevention' tier into the model by Scott *et al* to help minimise the impact of PSIs. This multilevel approach, firmly rooted in prevention reflects best practice in stress management more broadly,[43–45] which advocates minimising the stressors that compromise employee well-being (primary prevention), moderating individuals' reactions when exposed to stressors (secondary interventions) and providing individualised support to those experiencing strain (tertiary interventions).

However, numerous reviews to date examining what support is actually available for second victims (including scoping,[46 47] literature,[41 48] narrative,[42] integrative[39 49] and systematic)[50] have identified that such support is rare and limited. Even when support is available, various challenges affect whether and/or how it is used and received. These include a 'blame' culture,[49] staff resistance to usage, reluctance to ask for help, low awareness of availability[39 50] and a heavy reliance on volunteers due to lack of funding and resources, which can compromise availability and sustainability.[39 49 50]

Collectively, these reviews provide extensive coverage of the range of support interventions available internationally (>40 were documented in one review).[46] However, reviews to date have largely aimed to identify and describe second victim support interventions available, rather than to critically evaluate outcomes. Included papers have varied in nature, including descriptions of programme development and peer training workshops, explorations of risk managers' perceptions of second victim programmes and reports of implementation and/or usage data. While most reviews report on acceptability, perceived helpfulness, feasibility or implementation challenges, some[48–50] have also considered the intervention benefits or what has proven helpful or not helpful in their aims. However, only one study[47] cites the investigation of

| Table 1 | Four symptom groups associated with the 'second victim' experience |
|---------|----------------------------------------|
| **Symptom group** | **Individual symptoms** |
| Emotional | Shame; guilt; anxiety; anger; shock; depression; fear |
| Cognitive | Flashbacks; intrusive thoughts and nightmares; re-imagining; ruminating on details; helplessness; suicidal ideation |
| Physiological | Fatigue; sleep disturbance; nausea; headaches; gastrointestinal issues |
| Behavioural | Hypervigilance; withdrawal; irritability; social avoidance; emotional numbing; poor concentration; diminished memory; avoidance of triggers |

intervention outcomes for healthcare professionals and organisations as one of its aims. Searching the literature up to 2018, the scoping review by Wade et al[47] concluded that 'there is little scientific rigour from which to determine whether these programmes are indeed meeting their goals and objectives' (p. e70). As others have also noted,[39 46 48 50] we currently have limited understanding of what actually 'works' when it comes to supporting staff through PSIs.

There is broad agreement that supporting staff through PSIs is the responsibility of senior organisational leaders, and that such support should be comprehensive, part of a wider just, restorative patient safety culture, sustainable and well-funded.[39 41 42 46–50] However, for organisations to undertake such an investment and provide quality support, they require more robust evidence. There is a need to understand, comprehensively and systematically, what type of support staff actually want and what interventions can be provided to meet their desires (a key factor in the uptake of well-being interventions[51]) that are effective.

In this systematic review, we used a widely respected, multilevel stress prevention model[43] to map and synthesise the empirical evidence surrounding second victim support. Specifically, we aimed to answer the following research questions:

1. What support do healthcare staff want before and after a PSI?
2. What support is currently available to these staff?
3. How effective are existing support mechanisms/interventions designed to: (a) prevent poor psychological outcomes for healthcare staff after PSIs; (b) support staff in the aftermath of PSIs; (c) offer psychological support for those affected by involvement in PSIs.

## METHODS

### Design

We used systematic review with narrative synthesis[52] and followed the Preferred Reporting Items for Systematic Reviews and Meta-Analyses (PRISMA)[53] (for PRISMA checklist, see online supplemental file S1). The protocol was registered with the International Prospective Register of Systematic Reviews (PROSPERO; CRD42022325796).

### Patient and public involvement

No patients or members of the public were involved in the design, conduct or reporting of this review. However, we plan to disseminate our findings through our patient partner networks, which will be made available to people in a range of ways. Key messages will also be shared through social media.

### Eligibility criteria

Studies were eligible if they were empirical and provided information within any healthcare context regarding the support frontline healthcare staff desire before or after a PSI, OR the support currently available, OR the effectiveness of interventions aimed at preventing or mitigating

second victim phenomena. Searches were limited to studies published in English between January 2010 and December 2022. Our focus was much less on understanding the prevalence of workforce responses to error and investigations (the types of publications that emerged shortly after the term 'second victim' was coined by Albert Wu in 2000[23]) and much more on the types of interventions and their effectiveness. A number of key papers on staff support for PSIs were published in 2010, including the Institute for Healthcare Improvement's white paper on respectful management of PSIs,[54] the seminal paper by Scott et al[40] on organisational support for second victims and our original systematic review on coping with medical error.[55] To keep the review focused and manageable, we decided to limit our scope to publications since that time. Articles were excluded if they were grey literature, study protocols, opinion pieces, systematic reviews, meta-analyses, theses, focused on general staff well-being, stressful non-PSIs or aimed at reducing or increasing the reporting of medical errors.

### Study identification

Drawing on search terms used in our recent, related systematic review[31] and with support from a medical information specialist, synonyms and relevant concepts were developed for the two major concepts being explored: second victim and emotional support. A search strategy was developed iteratively and applied to the following electronic databases in June 2021 and updated in December 2022: Medline, Scopus, PubMed and Web of Science (for all search strategies, see online supplemental file S2). Reference lists of eligible articles were also searched and the Connected Papers software tool (https://www.connectedpapers.com/) was used to identify additional studies. Covidence systematic review software (Veritas Health Innovation, Melbourne, Australia) was used for study screening and management.

### Study selection and data extraction

After a preliminary scan of the eligible articles, we determined that a two-stage process was required for the review: stage 1 would address research questions 1 and 2; stage 2 to address research question 3. This represented a deviation from our published protocol. Using the eligibility criteria, four of the authors (RS, RL, RS-E, ES) with support from Dr Jane Heyhoe (JH), screened first titles and abstracts, then full texts for each potentially relevant study. Regular weekly meetings were held during this period to discuss screening decisions and disagreements, finalise eligible articles for inclusion and undertake a face validity check of the final set of studies. A PRISMA diagram was created to document the decisions made (figure 1).

Studies were selected if they could answer research questions 1, 2, 3 or a combination and coded within Covidence to denote inclusion at stage 1 or stage 2. All authors extracted the following information from the included studies: author, publication year, country, study

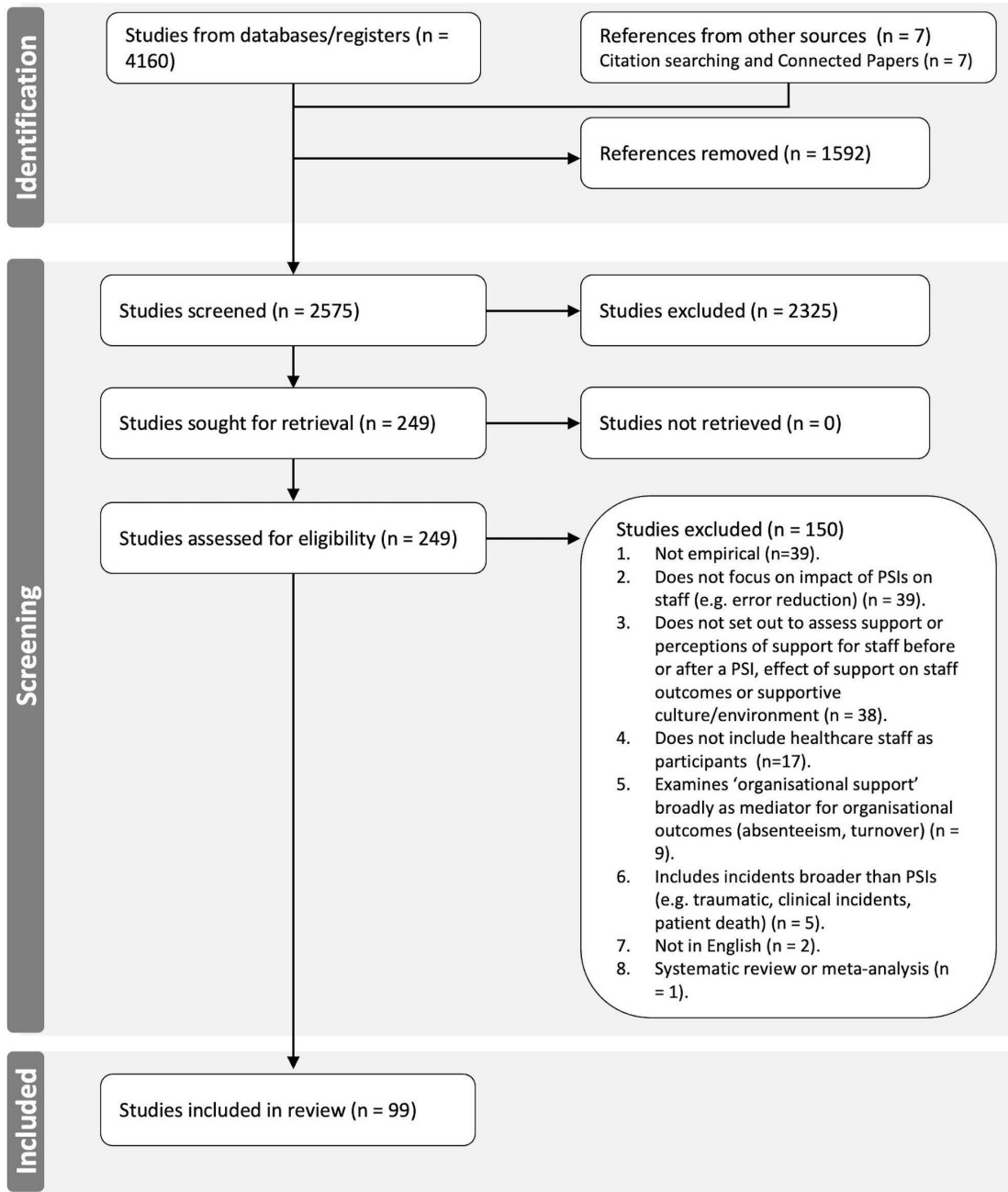

**Figure 1** PRISMA (Preferred Reporting Items for Systematic Reviews and Meta-Analyses) flow diagram of study selection process

design, setting, sample and key findings (qualitative and/or quantitative). In addition, all articles featuring individual interventions were categorised according to the level of organisational support offered, that is, primary, secondary or tertiary.

At stage 1, due to the large number of studies, we limited the extraction of 'key findings' to what healthcare staff said they wanted and what support they received. At stage 2, the extraction of 'key findings' was more comprehensive to allow us to examine effectiveness of specific interventions. Our stage 2 reporting is guided by the synthesis without meta-analysis (SWiM) in systematic reviews[56] framework (online supplemental SWiM checklist). We focused on outcomes, such as changes in confidence,

job satisfaction or burnout. We did not focus on feasibility or acceptability, therefore we did not extract any data regarding user acceptability and intervention usage. Additional 'intervention description' data were also extracted at stage 2, using the Template for Intervention Description and Replication (TIDieR) criteria.[57] At both stages, data extraction was conducted by all reviewers, with regular discussions to resolve disagreements. A random 20% sample of intervention effectiveness studies included in stage 2 was double checked independently by a second reviewer (JH).

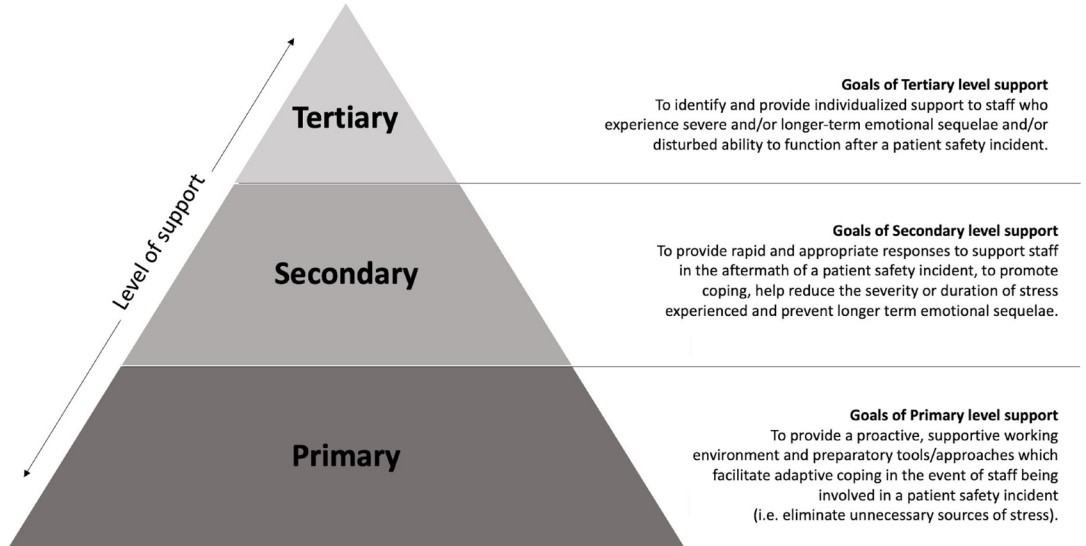

**Figure 2** A multi-level Organisational Staff Support Model (OSSM) for Patient Safety Incidents

## Assessment of study quality

Only studies included at stage 2 were assessed for methodological quality. This was conducted independently by two reviewers (RS, HH) using the 13-item quality assessment with diverse studies tool (QuADS),[58] selected for its appropriateness for study samples with different methodological approaches.

## Data synthesis

In this systematic review, we used an established multilevel stress prevention framework to help us organise and map the empirical evidence available. We adapted the intervention strategy by Cooper and Cartwright[43] originally designed for general workplace stress, to address the specific and unique workplace stressor/hazard of being involved in a PSI as a healthcare professional within a healthcare organisation (figure 2; for more details, see https://secondvictim.co.uk/supporting-second-victims/#why-support-second-victims).

Data at both stages were synthesised according to each research question using narrative synthesis. In addition, each form of support was coded according to the level it targeted: primary (ie, preventive, proactive), secondary (ie, aiding natural coping after an incident) or tertiary (ie, specialist one-to-one remedial support).

At stage 1, where a study reported support in ranked order of preference or most frequently received, we focused on the top three supports in each case. Where rankings were not available, such as in qualitative studies, we focused on the support most frequently reported as preferred and received. At stage 2, an initial description of the intervention effectiveness studies' results was produced, including descriptive statistics and an assessment of study quality. A narrative synthesis was conducted, where studies were organised to explore the effectiveness of available support in helping to prevent or alleviate the anticipated or actual psychological consequences of a PSI. Specifically, each intervention was coded as primary-level,

secondary-level or tertiary-level support to create three groups of studies. Each group was reviewed and synthesised separately, before completing a multilevel, holistic examination and synthesis. Regular meetings were held to discuss and refine emerging findings. Study information was systematically tabulated by: intervention name, description, study design and quality appraisal score, sample and setting, outcome measure related to effectiveness (eg, burnout, job satisfaction, confidence in coping), measure used and outcomes. Wherever available, we extracted p values as our standardised metric in examining intervention effects. Patterns within and between studies were explored to identify commonalities/differences and potential reasons for these.

## RESULTS

### Results of the search

We retrieved 4160 articles, 99 of which were included for data extraction (figure 1). All 99 studies were included at stage 1. Only 11 studies[59–69] presented outcome data and purported to assess effectiveness, therefore were subsequently also included in stage 2.

### Excluded studies

Studies were predominantly excluded at full-text review because they were not empirical (n=39), they did not focus on the impact of PSIs on staff (n=39) or they did not aim to assess staff support interventions for PSIs, healthcare staff perceptions of PSI support or the effect of PSI support on staff/organisational culture outcomes (n=38). Several exclusions were also made for studies that did not include frontline healthcare staff as participants (n=17), or examined 'organisational support' more broadly as a mediator for organisational outcomes (n=9).

### Characteristics of included studies

Included studies were primarily from the USA (n=41), Europe (n=33), East and South-East Asia (n=13) and Australasia (n=6). Studies were quantitative (n=52), qualitative (n=22) and mixed methods (n=25). The cross-sectional survey dominated (n=73); those with a qualitative component mostly used semi-structured interviews (n=23), focus/online discussion, action groups or both (n=10). Samples were most often multidisciplinary (n=53) with others featuring unidisciplinary groups (n=46), which were mostly of physicians (n=22), nurses (n=16). Specialties represented included obstetrics and gynaecology or midwifery (n=13), surgery (n=10), anaesthesiology (n=7), paediatrics (n=5) and intensive/critical care (n=5), psychiatry and mental health (n=3), pharmacy (n=1), radiotherapy (n=1), acute care (n=1), cardiology (n=1), urology (n=1) and internal medicine (n=1). Sample sizes ranged from eight participants to 3766 (based on n=92 studies: n=7 sample sizes were not clear).

### Stage 1: review findings

What support do staff want before and after a PSI? (Research question 1). What support is currently available to healthcare staff? (Research question 2).

Seventy-two studies investigated what staff wanted (n=15), received (n=20) or both (n=37). Thirty-five different forms of 'desired' and 11 different types of 'experienced' support were identified. Due to the scale of the data, a decision was made to focus on the top five most highly rated and experienced forms of support (table 2). However a detailed narrative summary of findings and references can be found in online supplemental file 4. The top three most desired and most experienced support are described below (see online supplemental file 5 for top four and five).

### *The top three most desired forms of support (research question 1)*

**#1 Peer support (n=28 studies):** peer support was the most desired form of support in every study that presented rankings. Specifically valued were peers sharing their own mistakes, checking in regularly and providing practical advice and reassurance about their professional capability to help restore confidence. However, the definition of 'peer support' differed between studies, from 'a respected peer to discuss the details of what happened', 'emotional support from colleagues', 'colleagues taking time to listen or help with workload' to 'conversations with peers'. Differences were also apparent in who second victims wanted peer support from. Some preferred 'informal peer support', from non-judgemental, empathic colleagues from their direct work environment, who fully understood the nature of the job and the incident itself. Others, wanted to speak with the whole team and some physicians wanted to talk only with other physicians.

**#2 Practical support and guidance (n=27 studies):** preferences for practical support after a PSI featured heavily across the sample, including rest and recuperation opportunities, temporary work adjustments and technical and procedural guidance. Having a specified peaceful location for recovery and regaining composure appeared as

**Table 2** Most desired compared with most experienced forms of support for coping with patient safety incidents

| Most desired | No of papers in sample (In Top 3 of surveys; In qual. or mixed methods papers) | Most experienced | No of papers in sample (In Top 3 of surveys; In qual. or mixed methods papers) |
|---|---|---|---|
| 1. Peer support | **N=28** *(n=16; n=12)* | 1. Peer support | **N=46** *(n=8; n=38)* |
| 2. Practical support and guidance | **N=27** *(n=7; n=20)* | 2. Managerial support | **N=23** *(n=5; n=18)* |
| 3. Professional mental health support | **N=21** *(n=8; n=13)* | 3. Post-incident debrief to clarify, process | **N=15** *(n=3; n=12)* |
| 4. Post-incident debrief to clarify, process | **N=18** *(n=8; n=10)* | 4. Professional mental health support | **N=11** *(n=0; n=11)* |
| 5. Supportive, learning culture | **N=14** *(n=1; n=13)* | 5. Mortality and Morbidity meeting | **N=7** *(n=3; n=4)* |

Primary-level support.
Secondary-level support.
Tertiary-level support.

a top three most desired form of support in the second highest number of studies. A desire for temporary job adaptations and accommodation was also evident, such as being freed from workloads, having a review of one's responsibilities and being allowed to complete tasks and responsibilities with a colleague. Time away from work was also commonly desired, or a period of leave to help distance oneself and recover from the event. All these were desires for staff to initiate themselves and discuss with managers, rather than being instructed by managers punitively due to loss of trust after a PSI. Numerous studies found that second victims wanted technical guidance and advice with procedures surrounding investigations, legal aspects and support in communicating with the affected patient or family.

**#3 Professional mental health support (n=21 studies):** specialist mental health support or counselling, including access to employee assistance programmes (EAPs), was highly desired: of the studies that ranked support strategies, counselling was rated in the top three by the joint third most studies. Staff across several studies wanted confidential counselling to be provided in-house by the organisation, while a similar number wanted professional help offsite, through EAPs. In some studies, healthcare staff wanted formal support by a mental health professional, while in others, staff wanted an opportunity to *access* this if needed. Ambivalence was also sometimes evident, with concerns about whether this would be acceptable and used, particularly by surgical teams.

### The top three most experienced forms of support (research question 2)

**#1 Peer support (n=46 studies):** peer support was the most frequently experienced form of support, ranking most common in studies that ranked support frequency. Healthcare staff gained support speaking to direct colleagues, especially those at a similar grade and those involved in the incident. However, some ambiguity surrounds the way peer support was discussed across the sample. Often insufficient detail is available to consistently identify and define peer support: some studies imply it is facilitated by the organisation in some structured, formal way; others imply it is more informal, talking to colleagues in one's team.

**#2 Managerial support (n=23 studies):** after peer support, support from supervisors, managers and department leaders was the next most frequently experienced form of support, according to included surveys. Qualitative and mixed methods studies also reported this extensively, detailing how managerial support helped to resolve emotional stress. Examples included giving staff space without chasing up incident reports, providing support via team meetings or one-to-ones and generally being proactive and accessible after an event. Managers' guidance and reassurance that they have confidence in individuals involved in an incident was highlighted as particularly important. Conversely, inadequate managerial support compounded isolation and hindered recovery.

**#3 Post-incident debrief (n=15):** debriefs were also commonly experienced by staff, however descriptions of what these entailed were not always provided. Where details were available, the format and implementation were inconsistent across studies. For example, qualitative studies highlighted that these might be voluntary, certain staff might not be involved or invited or the focus was on learning and attributing blame rather than emotional recovery.

### Discrepancies between support desired versus experienced

An overarching finding is that healthcare staff commonly experienced their organisation's support as insufficient, contingent on one's role/seniority, non-existent or unclear. Themes of resentment and injustice ran through many of the included studies. There appear to be discrepancies between what healthcare staff want and what they actually receive (table 2). 'Peer support' needs appear to be appropriately met overall, but overlooked or neglected is the very highly rated need for 'practical support and guidance'. Similarly, unmet is the desire for a supportive, learning-oriented culture; no experiences were reported of primary-level interventions. By contrast, the procedural, routine 'mortality and morbidity meeting' was not a top five most desired support, yet was the fifth most commonly experienced, despite not technically being a well-being intervention.

### Specific support interventions reported in empirical studies (research question 2)

In addition to the above 72 qualitative studies and surveys, a further 27 studies featured an evaluation of a single support intervention with healthcare staff. These are outlined in table 3, grouped according to whether they were primary-level or secondary-level interventions. No papers evaluating tertiary interventions were identified. Of these 27 intervention papers, 11 examined intervention effectiveness and were included at stage 2 of the review to address research question 3.

### Stage 2: review findings

How effective are existing support mechanisms/interventions designed to: (a) prevent poor psychological outcomes for healthcare staff after PSIs (primary prevention); (b) support staff in the aftermath of PSIs (secondary intervention); (c) offer psychological support for those affected by involvement in PSIs (tertiary intervention) (research question 3).

### Characteristics of studies included at stage 2

Eleven studies were included at this stage and are summarised in the online supplemental table. Of these 11 studies of intervention effectiveness, three featured primary-level interventions and eight involved secondary-level interventions. There were no papers of tertiary-level support. Most were conducted in the USA (n=7: 61, 62, 63–65, 67, 68), followed by Australia,[66] the UK,[59] Spain[60] and Indonesia.[69] Studies were quantitative (n=9: 60, 61, 63, 64, 65, 67–69) and mixed methods.[59 62] Samples

**Table 3** Individual interventions to support healthcare staff coping with patient safety incidents—reported in empirical research papers

| Support level | Intervention name description | Study reference | Setting; *healthcare staff sample* |
|---|---|---|---|
| Primary | The 'When Things Go Wrong' curriculum | 93 | University; *medical students* |
| | 'Interactive teaching session addressing personal and professional effects of medical errors' | 61 | University; *medical students* |
| | Adverse Events Committee Programme: annual 'Safety Education Workshop' | 94 | Hospital; *psychiatry residents* |
| | 'Resilience coaching intervention to prepare UK healthcare professionals for the occurrence of stressful events' | 59 95 | University, hospital; *multiple disciplines (qualified staff, trainees)* |
| | 'Mitigating impact in second victims' online programme | 60 | Online (hospital); *multiple disciplines* |
| Secondary | Adverse Events Committee Programme | 94 | Hospital; *psychiatry residents* |
| | Online, 10-chapter e-book reference guide for responding to and supporting those involved in an incident | 96 | Mental health organisation; *all staff* |
| | University of Missouri Healthcare's 'forYOU' peer support programme (wholly or modelled on, using three-tiered model of second victim support by Scott *et al*[40] | 40 63 66 67 69 97–101 | Hospital, including outpatients,[64 66] *OBGYN staff,*[67 101] *paediatric staff,*[63] *pharmacy staff,*[100] *neonatal ICU staff,*[99] *all staff*[40 69 97 98] |
| | The Brigham and Women's Hospital Peer Support Service | 102 103 | Hospital; *surgeons and trainees,*[102] *all staff*[103] |
| | The Resilience in Stressful Events programme | 62 65 104 105 | Hospital; *paediatric staff,*[104 105] *nurses*[62 65] |
| | 'Swaddle' second victim staff support programme (hybrid of above programmes) | 106 | Hospital; *all staff* |
| | Certified Registered Nurse Anaesthetist Peer Support Programme | 68 | Hospital; *certified registered nurse anaesthetists* |
| | The 'Buddy Study' peer support programme | 77 | Hospital; *OBGYN midwives; ER and internal medicine physicians* |
| Tertiary | None | | |

ER, emergency room; ICU, intensive care unit; OBGYN, obstetrics and gynaecology.

included unidisciplinary[59 61 62 65 68] and multidisciplinary groups in a specific department[63 67] or whole organisation.[60 64 66 69]

Intervention users were compared with non-users in around a third of studies.[62 66 67 69] A range of follow-ups was included in the sample: from a single survey pre-implementation and post-implementation[63 66 68] to multiple follow-ups (3 and 6 months post-intervention[67]; 3 and 9 months post-launch[64]; 1 and 2 years post-launch).[69] Two studies involved a single post-implementation survey.[62 65]

Three studies featured primary-level interventions.[59–61] The formats of these interventions are outlined in the online supplemental table. All aimed to raise awareness of the prevalence of adverse events, understanding of their personal and professional impact, to prepare participants for involvement in an adverse event and ensure they knew how to access resources and support after a PSI. In addition, Mira *et al*[60] targeted practical, personal preparation, including how to inform patients and what to say/not say to colleagues and Johnson *et al*[59] targeted

the development of participants' cognitive flexibility, self-esteem and positive attributional style. All primary-level intervention studies measured changes in users' knowledge (of PSIs, second victimhood and the impact, appropriate responses/actions in the aftermath[60]; of resilience, coping strategies, personal strengths[59]; of medical error leading to physician burnout[61]), two explicitly measured changes in confidence to cope with adverse events[59 61] and one measured resilience to cope with future events.[59] Changes in awareness of support resources and attitudes to error were also measured by one.[61] Purpose-designed questionnaires/tests were used in all cases, supplemented in one[59] by the Brief Resilience Scale.[70]

Eight studies featured secondary-level interventions,[62–69] all peer support programmes which aimed to offer tailored, flexible support to staff in the aftermath of an adverse event, with referral to more specialist psychological support if required/desired (for formats, see online supplemental table). The scope varies between programmes. One programme[66] was reported as part of a wider just culture organisational change initiative.

Specific programme goals differed, including 'to provide timely support' after a stressful event,[65] 'to decrease second victim distress'[68] and 'to mitigate challenges and improve the well-being of staff' immediately after an incident, and, long-term, enhancing resilience, self-care and 'connections with colleagues and the organisation'.[66] Five papers[63 64 66 67 69] feature programmes based on or informed by the University of Missouri Healthcare 'forYOU' framework (grounded in three-tiered interventional model of support by Scott *et al*), which aims to promote the healthcare professional's transition through emotional recovery after an incident[40] (p. 238). Some included programmes[63 64 66] have slightly adapted this forYOU model to suit their local needs (online supplemental table). Two papers relate to the Resilience in Stressful Events (RISE) programme,[62 65] and the final study features the Clinical Registered Nurse Anaesthetist peer support programme.[68]

In terms of outcomes measures used in secondary-level intervention studies, four studies[63 64 66 68] measured changes in levels of psychological and physical distress, perceptions of colleague, supervisor and institutional support, professional self-efficacy, absenteeism and turnover intentions, using the Second Victim Experience and Support Tool (SVEST).[71] One study[67] measured changes in current stage of recovery after a PSI based on the six-stage second victim recovery trajectory by Scott *et al*[7]. It is unclear whether this team used an existing or purpose-designed tool. Differences in burnout, job satisfaction and resilience were investigated in one study,[62] using a combination of existing measures (abridged Maslach Burnout Inventory,[72] Job Satisfaction Scale,[73] Brief Resilience Scale).[70] Two studies examined organisational-level data: changes in perceptions of organisational culture post-implementation,[69] using the Hospital Survey on Patient Safety Culture (HSOPSC)[74] and costs associated with implementing versus not implementing the programme, through modelling probable savings from reduced sickness absence and turnover.[65]

### Study quality

Study quality was variable. Insufficient detail was an overarching issue across these intervention studies, particularly with regard to enabling study replication. Theoretical underpinning was limited (most studies only referred to second victimhood in their introduction) and insufficient description tended to be provided of samples and sampling strategies in relation to aims. There was only one randomised trial in the sample, meaning that most studies did not use a comparison group and that staff/worksites were assigned non-randomly to interventions when testing for effectiveness. These are sources of potential bias. Self-reported data were relied on in the majority of cases, increasing the potential for response bias. Evidence of stakeholder engagement in design was limited across studies, although most seemed to be led by clinical teams. We did not exclude any study based on its quality appraisal outcome.

Our TIDieR assessment of the intervention descriptions also revealed that theoretical underpinnings were provided in less than half of studies included at stage 2. Details regarding intervention materials, associated procedures/activities and facilitators were well described overall. Just one study explicitly mentioned fidelity assessment of intervention delivery, although a small number of studies referenced regular facilitator meetings for supervision, debriefing, learning and continuous improvement.

### How effective are existing primary-level (preventive) interventions?

Evidence was largely positive for the effectiveness of the three primary-level interventions in preparing healthcare staff and trainees for exposure to adverse events. All three studies reported significant improvements in targeted outcomes, particularly in relation to knowledge and awareness of resources in relation to the second victim phenomena online supplemental table. Users of the 'Mitigating Impact in Second Victims' online training programme by Mira *et al*[60] demonstrated significant improvements in their knowledge of basic patient safety concepts, the prevalence and nature of adverse events. Similarly, medical students participating in a 1-hour teaching session training by Musunur *et al*[61] demonstrated significant improvements in confidence in their ability to recognise and cope with medical error and in their awareness of resources to cope with and report medical error. Participants in a half day workshop by Johnson *et al*[59] also showed significant improvements in confidence with coping with adverse events, knowledge about resilience factors, coping strategies and awareness of personal strengths, which were maintained 4–6 weeks after the final workshop.

### How effective are existing secondary-level (responsive) interventions?

Evidence for the effectiveness of secondary-level peer support-based interventions was more mixed. With regard to adaptations of the forYOU programme, Klatt *et al*[64] identified no significant changes on any SVEST dimension ratings when following up at 3 and 9 months after their programme launch. In the only randomised trial in the sample, Rivera-Chiauzzi *et al*[67] found no significant differences at 6 months post-enrolment between their intervention and control groups regarding the ability to recover from an adverse event and reach the optimal 'thriving' recovery stage; both groups of staff recovered equally well.

More positive findings were reported, however, in studies by Finney *et al*[63] and Wijaya *et al.*[69] Finney *et al*[63] found significant changes in the institutional support and turnover intentions dimensions in SVEST: they report that, post-implementation, 'inadequate' ratings of institutional support and turnover intentions fell significantly (however, p values in the article body text (p=0.014—significant) and table 2 (p=0.09—not significant) do not correspond). Wijaya *et al*[69] reported significant increases in HSOPSC ratings in the intervention

hospital 1 year after introducing the programme, which were significantly higher than those of staff in the control hospital. This improved safety culture was sustained 2 years post-introduction of the programme. Morris et al[66] found no significant differences between 'Always There' programme users and non-users 1 year post-launch on any overall dimension of SVEST, although users did rate two (out of three) items in the 'institutional support' dimension significantly higher than non-users.

Turning to the RISE programme, two studies evaluated its impact specifically on the nursing workforce in their hospital. Moran et al[65] calculated that having the RISE programme could save a hospital US$1.81 million per year in reducing turnover and absence. Connors et al[62] found no significant difference in job satisfaction between either nurse users and non-users or between nurse leader activators and non-activators. Compared with non-users, nurses using RISE were significantly more likely to report higher levels of resilience and some level of burnout. Nurse leaders who did not implement RISE for their staff reported significantly higher levels of resilience compared with those who did.

Finally, a quality improvement project by Thompson et al[68] found no statistically significant differences between pre-intervention and post-intervention implementation SVEST scores of all departmental certified registered nurse anaesthetists surveyed, 1 month after implementing their peer support programme.

## DISCUSSION

We adapted and used a widely respected, multilevel stress prevention model[43] to map the empirical evidence for what type of support healthcare professionals *want* for PSIs, what they tend to *receive* and what organisational strategies are *effective*. Our findings revealed that healthcare professionals want organisational support at multiple levels: a supportive and learning oriented culture (primary prevention level support), peer support, practical support and guidance, post-incident debriefing to clarify and process what happened (secondary level support) and access to professional mental health support (tertiary level support). However, there is limited evidence that organisations are giving staff what they want: overwhelmingly, organisational support for PSIs is perceived as inadequate, non-existent or dependent on one's organisational status. Where support *is* provided, secondary-level approaches dominate—largely at a local level from peers and one's line manager. There is a gap in primary-level (preparatory, proactive, preventive) approaches, which tend to be missing from staff reports.

For healthcare organisations seeking to invest in evidence-based interventions to support staff with PSIs, effectiveness data are lacking. There is more robust evidence for the effectiveness of *primary interventions* designed to equip and prepare staff for involvement in PSIs, than for *secondary interventions*, which aim to mobilise support and promote staff coping *after* involvement in a PSI. However, despite this, secondary-level interventions dominate the literature, particularly the popular peer support programmes, which have mixed evidence regarding their effectiveness. No studies were identified featuring a tertiary intervention, that is, individualised support for staff experiencing severe and/or long-term emotional sequelae and/or disturbed ability to function after a PSI. While there is compelling evidence more broadly that cognitive behavioural therapy and eye movement desensitisation and reprocessing are effective in treating PTSD,[75] data are not available to be able to recommend their particular effectiveness in alleviating second victim-related sequelae.

Our stage 1 findings regarding what staff *want* and what they *receive* reflect those of previous reviews. Their overall desire for comprehensive support, within a just culture, to be able to navigate PSIs adaptively is consistent with the conclusion by Seys et al[42] that organisational action plans need to incorporate preventive strategies, peer support and professional and clinical support. Our finding that staff rarely *receive* formal organisational support is also consistent with previous reviews,[39 48 50] and a recent scoping review examining support for surgeons[76]: support tends to be 'piecemeal' and informal, with staff mostly relying solely on the good will and support of their clinical colleagues in the aftermath.

Our stage 2 finding that there is a lack of robust empirical evidence for the effectiveness of specific PSI-related interventions is also in line with previous research.[39 46–48 50] As others have noted, this may be attributable, in part, to being a relatively young research field and an initial focus on intervention feasibility, acceptability and implementation challenges.[47 77] We identified only one RCT. Few included a theoretical underpinning or basis for their design. Organisational-level outcome data (retention, sickness absence, cost-effectiveness, organisational culture) featured in just two studies.[65 69] It is becoming increasingly important for organisations, particularly post-COVID-19, to have reliable intelligence about how best to support staff after PSIs and what works, to justify investment to stakeholders including staff. Therefore, there is a need for the field to evolve rapidly and for robust studies to be funded.

Collectively, the wide range of healthcare staff voices across our included studies corroborate the need for organisational support for PSIs to be multilevelled and comprehensive. This finding supports previous reviews of second victim support,[41 42 48–50] and reflects the wider occupational stress literature more generally.[43–45 78] A lack of primary-level interventions in healthcare organisations to help staff with PSIs is perhaps not surprising: organisations generally tend to prefer secondary and tertiary interventions.[43] Targeting or changing individuals is considered easier, more cost-effective, tangible/visible and less disruptive than changing organisations, for example, changing organisational culture, working conditions, job designs[43] (p. 10). Indeed, it is noteworthy that the interventions and approaches within our sample

tended to target the individual, rather than focusing on teams (where incidents happen and must be dealt with). As Busch *et al*[50] surmise in their systematic review of second victim support interventions, the emphasis appears to be on reducing distress after a PSI, trying to 'fix what has gone wrong in the healthcare provider' (p. 12).

However, disregarding primary-level support may be problematic for two reasons. First, research surrounding broader work-stress interventions suggests that primary interventions may be more effective at stress reduction than secondary or tertiary approaches.[79 80] Second, multi-level intervention strategies combining individual-level and organisational-level interventions are more effective than uni-level interventions alone,[79 81 82] therefore organisations risk poorer returns on their investment if they fail to adopt a comprehensive approach. Primary-level interventions, particularly those addressing improvements in culture and working practices, appear to be especially important in healthcare settings, because of the significant impact the presiding patient safety culture has on staff's recovery trajectory after a PSI.[16] Indeed, previous reviews have repeatedly underlined that a systemic approach to patient safety and a just, restorative culture are essential foundations for the provision of quality staff support for PSIs.[39 41 42 46 50] A recent scoping review of factors underpinning the effective implementation of interventions to support staff with PSIs (and other stressful clinical events) also identified that an organisational culture based on trust, non-judgement and multidisciplinary action is vital for success.[83]

Organisations that fail to invest in primary-level support—specifically, implementing a fair and just organisational culture—but provide a formal peer support programme (secondary level) or EAP (tertiary level), risk staff viewing these provisions as 'tokenistic', or as signifiers that the organisation has abdicated responsibility for its staff because it is not tackling poor culture which is a major source of distress.[84] In fact, when employers ask their staff to identify factors that would have the greatest impact on their well-being at work, they overwhelmingly suggest better communication, staffing and interpersonal connectivity, rather than issues targeted by well-being programmes.[39] Themes of injustice and resentment were also apparent within many included studies in the review, which, independent of involvement in a PSI, are risk factors for turnover intentions in healthcare staff.[85]

In line with previous research, our review found that peer support is a highly desired and commonly received form of support.[15 31 40 49 50] However, we also identified considerable ambiguity around the term 'peer support' within included studies: it was often unclear whether it referred to informal support from team colleagues or a formal peer support programme. Given that formal peer support programmes are the most common organisational intervention provided, this has highlighted a need to ensure that such programmes are meeting the right requirement, particularly since evidence for effectiveness remains elusive.[47] For example, if a model offers peer support from outside one's team, but one's home team is unsupportive and one's leader is critical, then ongoing, day-to-day recovery in that location is likely to be more difficult.

Perhaps unsurprisingly, manager support emerged from this review as the second most commonly received form of organisational support, although there were also numerous accounts of inadequate managerial support compounding feelings of isolation and hindering recovery. This was also found in our related review.[31] The way in which supervisors or leaders treat their staff is crucially important, because, to staff, they 'embody' the organisation: staff form perceptions and experiences of their organisation from their supervisors/leaders behaviour and treatment of them.[86] Leadership quality has long been associated in the field of occupational health psychology with both positive outcomes, such as staff well-being, psychological safety and negative outcomes, including stress, heart disease and workplace incidents.[80] In healthcare in particular, compassionate leadership is being increasingly linked to staff well-being and positive safety culture.[87 88] The vital contribution leaders make to the implementation success (or failure) of organisational support programmes for PSIs was also highlighted recently by Guerra-Paiva *et al*.[83] Given managers' pivotal roles in supporting staff with PSIs, there is a strong case for organisations to provide compassionate leadership development in this area within their multilevel PSI support strategy, as a proactive, preventive, primary-level occupational health psychology intervention.[80]

Of significance is that the vast majority of intervention developers and paper authors within our stage 2 sample are clinicians. Secondary-level interventions dominate; fewer studies address primary-level prevention of second victimhood. This suggests that most interventions are pragmatic, responsive solutions to meet needs recognised at ground level by clinicians themselves. It also suggests that the recognition of and ownership for second victimhood as a serious, intrinsic occupational health risk to be addressed is located in clinical or patient safety teams; it does not reside with organisational senior leadership teams. However, as other reviewers have identified,[41 42 49 50] building the much-needed comprehensive multilevel strategies requires commitment from senior leadership teams of resources, funding and time; it cannot solely be responsibility of patient safety teams or local clinical departments.

### Strengths and limitations of this study

To our knowledge, this is the first systematic review to apply an established, occupational health stress prevention model to examine second victim requirements and organisational support solutions. This model is likely to be familiar to organisational development, human resources and occupational health teams in healthcare organisations, which we hope will aid dissemination and consideration at senior organisational levels. A second strength

of the review is that it specifically addressed intervention effectiveness.

However, a number of limitations must be acknowledged. First, despite including a medical librarian on our review team, it is possible that our search strategy did not locate all the relevant studies, due to focusing on papers in English, the broad scope of the area and outcomes or our selection and number of bibliographic search engines. As other reviewers have noted,[47] nomenclature varies significantly within this literature, therefore we may have missed important papers, which featured more unusual or unspecific terms. For these reasons, we may also have excluded relevant papers at the title and abstract screening stage, particularly those featuring primary prevention approaches, because these may have had broader goals and included many different types of organisational process.

Second, while starting our searches from 2010 aided the review focus and manageability, it may have omitted relevant older papers, and since our last update in December 2022, other relevant papers may have been published that we have not captured. We are aware of two recent empirical papers related to the Austrian 'KOHi' collegial help programme,[89 90] which is modelled on RISE and forYOU. However, neither paper reports outcome/effectiveness data, therefore would not have been included in stage 2 of our review.

A third limitation is that some methodological steps we undertook are known to be prone to subjective judgement and lack of transparency, particularly data extraction, quality assessment and narrative synthesis. To address this, screening, data extraction and assessment of methodological quality were conducted independently by two reviewers, and 20% of completed data extraction templates were independently double checked by a second reviewer. Regular meetings were held throughout to discuss any resolve disagreements, to review and refine emerging findings and outcomes documented to provide an audit trail of our decisions. Our reporting of stage 2 findings was guided by the SWiM framework.

Finally, our sample included studies that were heterogeneous in terms of design, sample sizes, data description and analysis, outcomes measured. A high proportion of papers featured cross-sectional data collected through self-report questionnaires, which are susceptible to non-response and recall bias. Study quality was also variable. As other reviewers have highlighted,[47 50] these factors currently render robust meta-analyses of interventions impossible, and there is a need for the field to agree on consensus on appropriate outcome measures and tools.

### Practical implications

The findings from this systematic review provide compelling evidence that healthcare professionals are routinely exposed to compounded harm after PSIs because of a lack of institutional-level responsibility and support. A quality PSI support programme should be an integral part of every healthcare organisation's strategy, policies and practices, rather than operating through the goodwill of passionate volunteers. Such programmes require the full support of senior leadership teams, with strong investment and promotion for sustainability.[41 49 50 83] Providing quality support to healthcare professionals for PSIs promotes adaptive recovery and has the potential to reduce defensive practice, promote healing, open up possibilities for restorative approaches and importantly, retain staff at a time when there are shortages of trained healthcare professionals worldwide.

This review provides organisations with robust, evidence-based data on what staff want, where there are currently gaps and what works—to inform the development of PSI support initiatives. It is hoped that presenting the findings through a familiar and respected organisational stress management framework may support a wider dialogue of the need to provide quality support for healthcare professionals, beyond patient safety and clinical teams to senior leadership teams, including organisational development, human resources, occupational health and educational leaders.

### CONCLUSION

Despite 20 years of investment to reduce the number of PSIs, they continue to persist[91]: 'to err is human'.[92] Given the fact that healthcare organisations can never entirely remove this intrinsic hazard, the importance of having an effective, comprehensive institutional support strategy in place for PSIs is evident, to aid adaptive recovery, minimise distress, sickness absence and turnover.[6] There is an incontrovertible, pressing need for healthcare organisations to address the multifaceted occupational health risk for their staff of being involved in a PSI. Organisations need reliable evidence about what interventions are effective, however this is currently lacking. In order to advance the field, we require more methodologically robust studies, which describe theoretical bases for interventions and involve in their design comparison groups, randomisation, organisational-level and individual-level outcomes. In addition, more research is needed into primary-level and tertiary-level interventions to help healthcare professionals with PSIs.

**Acknowledgements** We extend our sincere thanks to Dr Jane Heyhoe for her support of the team during the course of this research, prior to her departure from academia, and to Stephen Mears of the University of New South Wales, Australia, for his assistance with the database searches.

**Contributors** Conceived and designed the experiments: RS-E, RH, RS, ES, HH, RL. Performed the experiments: RS-E, RH, RS, ES, HH, RL. Analysed the data: RS-E, RH, RS, ES, HH, MM-E, RL. Contributed to materials/analysis tools: RS-E, RH, RS, ES, HH, MM-E, RL. Wrote the paper: RS-E, RL, RH. Author acting as guarantor: RS-E.

**Funding** This study was funded by the National Institute for Health and Care Research (NIHR) Yorkshire and Humber Patient Safety Translational Research Centre and the Patient Safety Research Collaboration

**Disclaimer** The views expressed are those of the authors, and not necessarily those of the NIHR or the Department of Health and Social Care.

**Competing interests** All authors have completed the ICMJE uniform disclosure form at http://www.icmje.org/disclosure-of-interest/ and declare the following: all

authors (except RH) were supported by the National Institute for Health and Care Research (NIHR) Yorkshire and Humber Patient Safety Translational Research Centre (PSTRC-2016-006) and the NIHR Yorkshire and Humber Patient Safety Research Collaboration. The authors declare no financial relationships with any organisations that might have an interest in the submitted work in the past 3 years, and no other relationships or activities that could appear to have influenced the submitted work.

**Patient and public involvement**  Patients and/or the public were not involved in the design, or conduct, or reporting, or dissemination plans of this research.

**Patient consent for publication**  Not applicable.

**Ethics approval**  Not applicable.

**Provenance and peer review**  Not commissioned; externally peer reviewed.

**Data availability statement**  All data relevant to the study are included in the article or uploaded as supplementary information. Not applicable.

**ORCID iDs**
Ruth Simms-Ellis http://orcid.org/0000-0002-1010-6648
Reema Harrison http://orcid.org/0000-0002-8609-9827
Rebecca Lawton http://orcid.org/0000-0002-5832-402X

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
