## [Reviewer comments · BMJ Open]

ARTICLE DETAILS

Title (Provisional)

Avoiding 'second victims' in healthcare: What support do staff want for coping with patient safety incidents, what do they get, and is it effective? A systematic review.

Authors

Simms-Ellis, Ruth; Harrison, Reema; Sattar, Raabia; Sweeting, Elizabeth; Hartley, Hannah; Morys-Edge, Matthew; Lawton, Rebecca /

VERSION 1 - REVIEW

Reviewer	1
Name	Mira, José
Affiliation	Miguel Hernandez University of Elche, Health Psychology
Date	25-Jun-2024
COI	None

Timely, well-constructed, and well-directed study. The methodological aspects are carefully considered, and the authors provide sufficient detail to allow for replication. There are just a few points that the authors may want to consider if they believe it would help the study:

1. Justify the selection of the study years.
2. The assertion that no other systematic reviews are known could be reconsidered; several reviews and scoping reviews on similar questions have recently been published. There are systematic, but are many common aspects
3. In this regard, the discussion could be enriched by comparing the findings with those of other authors.
4. No information is provided on actions with teams; everything seems focused on individuals. This aspect deserves to be commented on.
5. There are some examples of interventions, such as Austria experience, that are not cited.
6. Both RISE and other programs address second victims of highly stressful events not only related to adverse events. This aspect, among others, has prompted a recent review of the second victim concept.

7. Limitations are cited (2), but the authors could be a bit more self-critical in this regard.

8. The practical implications could be highlighted more clearly.

I find this to be a well-done, relevant study and highly recommend its publication.

Reviewer	2
Name	Mukherjee, Shatavisa
Affiliation	School of Tropical Medicine, West Bengal
Date	04-Jul-2024
COI	I declare having no competing interests.

Well presented.

VERSION 1 - AUTHOR RESPONSE

Reviewer #1 Comments to the Author

Timely, well-constructed, and well-directed study. The methodological aspects are carefully considered, and the authors provide sufficient detail to allow for replication. I find this to be a well-done, relevant study and highly recommend its publication.

We thank Reviewer 1 for their positive feedback, and for their constructive and collaboratively phrased suggestions.

1. Justify the selection of the study years.

We believe we have now done this. Searches were limited to studies published in English between 2010 and the end of 2022. Our focus was much less on understanding the prevalence of workforce responses to error and investigations (the types of publications that emerged shortly after the term 'second victim' was coined by Albert Wu in 2000) and much more on the types of interventions and their effectiveness. A number of key papers on staff support for PSIs were published in 2010, including the Institute for Healthcare Improvement's white paper on respectful management of PSIs, Scott et al.'s seminal paper on organisational support for second victims and our original systematic review on coping with medical error. Therefore, to make the review focused and manageable, we chose to focus on publications since this time.

2. The assertion that no other systematic reviews are known could be reconsidered; several reviews and scoping reviews on similar questions have recently been published. There are systematic, but are many common aspects

Thank you – on reflection we understand what you mean. We have now improved the introduction to be more measured in this regard, synthesise and highlight more clearly the key findings from previous reviews. We now include challenges affecting whether and/or how support is used and received, including: a 'blame' culture, staff resistance to usage, reluctance to ask for help, low awareness of availability, a heavy reliance on volunteers due to lack of funding and resources, which can compromise availability and sustainability, with references. We also acknowledge previous findings that there is broad agreement that supporting staff with PSIs is the responsibility of senior organisational leaders, that support should be comprehensive, part of a wider just, restorative patient safety culture, sustainable and well-funded.

3. In this regard, the discussion could be enriched by comparing the findings with those of other authors.

We have integrated more findings from previous reviews into the discussion to enrich the discussion and acknowledge more clearly how our findings support and extend previous studies. We also identified two additional reviews, and have incorporated these:

- Guerra-Paiva et al. Key factors for effective implementation of healthcare workers support interventions after patient safety incidents in organisations: a scoping review. *BMJ Open* 2023:13.
- Chong et al. Scoping review of the second victim syndrome among surgeons: understanding the impact, responses, and support systems. *American Journal of Surgery* 2024:229.

4. No information is provided on actions with teams; everything seems focused on individuals. This aspect deserves to be commented on.

Thank you. This is an interesting point. We have now commented on this focus on individual-level interventions in the discussion, particularly where we reflect on a lack of primary prevention and organisational level interventions. This point fits well here.

5. There are some examples of interventions, such as Austria experience, that are not cited.

Thank you for highlighting this. We believe this is the KOHI intervention, based on RISE and forYOU. We identified two empirical publications – one describing peers experiences of training (2024) and one (under review - preprint) of users' experiences. We have incorporated this into our limitations. We missed it as an intervention due to our last search being 2022, but we have commented that it is based on existing programme models we have discussed and would not have been eligible for inclusion at Stage 2 of our review because it does not report outcome data.

6. Both RISE and other programs address second victims of highly stressful events not only related to adverse events. This aspect, among others, has prompted a recent review of the second victim concept.

Thank you for raising this interesting point. We have added a sentence into page 25 of the manuscript – which deals with the peer support programmes - “the scope varies between programmes” – to encompass the fact that some also address stressful clinical events more widely.

7. Limitations are cited (2), but the authors could be a bit more self-critical in this regard.

Thank you. We agree we needed to be more self-critical! This was also raised by the Associate Editor, and has now hopefully been addressed. Please see the earlier section in our response to the Associate Editor.

8. The practical implications could be highlighted more clearly.

We have now included practical implications as a new subheading within the discussion, which notes:

- The compelling evidence that healthcare professionals are routinely exposed to compounded harm after PSIs because of a lack of institutional-level responsibility and support.
- A quality PSI support programme should be an integral part of every healthcare organisation's strategy, policies and practices, rather than operating through the goodwill of passionate volunteers.
- Such programmes require the full support of senior leadership teams, with strong investment and promotion for sustainability.
- Providing quality support to healthcare professionals for PSIs not only promotes adaptive recovery: it also has the potential to reduce defensive practice, promote healing, opens up possibilities for restorative approaches, and importantly, retain staff at a time when there are shortages of trained healthcare professionals worldwide.

- This review provides organisations with robust, evidence-based data on what staff want, where there are currently gaps and what works – to inform the development of PSI support initiatives.
- It is hoped that presenting the findings through a familiar and respected organisational stress management framework may support a wider dialogue of the need to provide quality support for healthcare professionals, beyond patient safety and clinical teams to senior leadership teams, including Organisational Development, Human Resources, Occupational Health and Educational leaders.

Reviewer #2 Comments to the Author

Well presented.

We thank reviewer 2 for their comments.

VERSION 2 - REVIEW

Reviewer	1
Name	Mira, José
Affiliation	Miguel Hernandez University of Elche, Health Psychology
Date	20-Dec-2024
COI	

Changes and explanations are very appreciate